# Taxonomic Uncertainty and Its Conservation Implications in Management, a Case from *Pyrus hopeiensis* (Rosaceae)

**Xian-Yun Mu** [1,*] , **Jiang Wu** [2] **and Jun Wu** [3]

1. College of Ecology and Nature Conservation, Beijing Forestry University, Beijing 100083, China
2. College of Biological Sciences and Technology, Beijing Forestry University, Beijing 100083, China; wujiang9212@163.com
3. Centre of Pear Engineering Technology Research, State Key Laboratory of Crop Genetics and Germplasm Enhancement, Nanjing Agricultural University, Nanjing 210095, China; wujun@njau.edu.cn
* Correspondence: xymu85@bjfu.edu.cn

**Abstract:** Improved taxonomies and phylogenies are essential for understanding the evolution of organisms, the development of conservation plans, and the allocation of funds and resources, especially for threatened species with uncertain identities. Pears are an economically and nutritionally important fruit, and wild pear species are highly valued and protected because of their utility for the development of cultivars. *Pyrus hopeiensis* is an endangered species endemic to North China, which is sympatric with and difficult to distinguish from the widely distributed and morphologically similar species *P. ussuriensis*. To clarify its taxonomic identity, principal coordinate analysis was performed using 14 quantitative and qualitative characters from *P. hopeiensis*, *P. ussuriensis*, and *P. phaeocarpa*, and phylogenomic analysis was performed based on whole-genome resequencing and whole plastome data. *Pyrus hopeiensis* was synonymized with *P. ussuriensis* based on morphological and phylogenetic evidence, as well as our long-term field studies. *Pyrus hopeiensis* is proposed to be excluded from the list of local key protected wild plants. Given that the holotype of *P. ussuriensis* was not designated, a lectotype was designated in this work. Integrative evidence-based taxonomic study including museomics is suggested for organisms with uncertain identities, which will contribute to biodiversity conservation.

**Keywords:** endangered species; integrative evidence; *Pyrus hopeiensis*; *Pyrus ussuriensis*; taxonomic uncertainty; biodiversity conservation

## 1. Introduction

The intensification of human activities, especially since the Industrial Revolution, has negatively affected global climate and biodiversity and is responsible for the Sixth Mass Extinction [1]. Rare and endangered plants are important components of biodiversity, and many of them are facing high extinction pressure because of inadequate conservation management [2], as well as taxonomic uncertainty [3,4]. Accurately identifying plants is a huge challenge, and taxonomic stability is an elusive goal [5,6]. Although studies of the evolution and taxonomic identity of species can greatly contribute to our knowledge of the organisms that we are interested in protecting, taxonomic problems might occur because of improvements in our knowledge of the phylogeny and evolution of organisms, as well as the recognition of previously made nomenclatural errors [7]. More than half of economically important tropical African ginger specimens from 40 herbaria in 21 countries are likely to be wrongly named [8], which can affect the conservation status assessment of related species. *Haematocarpus subpeltatus* Merr. (Menispermaceae), a misidentified critically endangered species that has received much conservation focus, has been identified as a new record in China, but the real entity that should be given priority for conservation, *Eleutharrhena macrocarpa* (Diels) Forman, has been clarified by molecular phylogenetic analyses [9]. These

findings highlight the significance of taxonomy and molecular phylogeny in biodiversity conservation, especially for species with taxonomic uncertainty.

Pears are an economically and nutritionally important temperate fruit that have been cultivated for more than 3000 years [10]. The genus *Pyrus* L. (Rosaceae) is geographically divided into oriental and occidental pears [11]. Considerable morphological variation and extensive hybridization have been documented in the genus [12,13]. The number of species recognized in *Pyrus* varies from 21 to more than 80 [11,14], and 73 of them are abundant in Eurasia [15]. Although remarkable advances have been made in the genetics and breeding of *Pyrus* in recent years, and whole genomes of several species have been published, e.g., [10,13,16–18], the taxonomy, diversification and phylogeny of *Pyrus* remain unclear, several currently accepted species such as *P. caucasica* Fed., *P. pyraster* (L.) Burgsd. and *P. spinosa* Forssk. are not monophyletic in the phylogenetic trees [12,19], suggesting the necessity of further taxonomic and phylogenetic studies. One reason for the lack of robustness of the molecular trees of *Pyrus* in previous studies is the few numbers of molecular markers employed. Whole nuclear genome sequences can provide vital information for reconstruction species' phylogenies [20], and complete plastomes can also provide key information for phylogenetic studies as well as for detecting hybridization events [21,22]. A hybrid usually neighbors one of its parents in the nuclear tree and group with another parent in the plastome tree, and phylogenetic discordance (cytonuclear discordance) is usually detected for hybrid species. Hence, extensive phylogenomic analyses are necessary for advancing our understanding of the phylogeny of *Pyrus*. This will shed light on the taxonomy and evolution of pear species and enhance pear production.

China is a diversity center for oriental pears, as 14 native species, including 5 primary wild species and over 2000 cultivars, have been reported to date [13,23,24]. Among them, *Pyrus hopeiensis* Yü, a taxonomically controversial species endemic to North China, is thought to be a hybrid of *P. ussuriensis* Maxim. and *P. phaeocarpa* Rehd. [25]. It is classified as an endangered species and is listed as a key protected wild plant of Hebei Province; it has thus received much conservation attention, e.g., [26–29]. *Pyrus hopeiensis* can be distinguished from *P. phaeocarpa* from several morphological characteristics, such as its persistent calyx and spiny serrate leaf margin (vs. caducous calyx and serrate leaf margin in *P. phaeocarpa*). However, differentiating *P. hopeiensis* and *P. ussuriensis* is more challenging. *Pyrus ussuriensis* is an important germplasm resource that is widely distributed in northern China, the Russian Far East, North Korea, and Japan, and more than 150 cultivars have been obtained from this species in China [30]. *Pyrus hopeiensis* has been documented in Beijing, Hebei, and Shandong provinces, and its type locality is Jieshi Mountain, a small hill with an altitude of 695 m in Changli County, Hebei Province, which is in the eastside of Yan Mountain in North China [23,25,31]. Based on the original description [25], *P. hopeiensis* can be distinguished from *P. ussuriensis* by its brown fruit and obvious spots on its surface (vs. yellow fruit with fewer spots). However, numerous individuals with intermediate morphological characteristics have been identified in the field and herbaria. Taxonomic ambiguity should be clarified, as this would aid conservation efforts for endangered and regional protected species. Hence, further morphological and phylogenomic studies should be performed to unravel the identity of *P. hopeiensis*.

In the present study, we aim to: (1) assess morphological differences among *P. hopeiensis*, *P. phaeocarpa*, and *P. ussuriensis*, (2) reconstruct the molecular phylogenetic tree and test the hybrid identity of *P. hopeiensis* based on whole genome resequencing data, and (3) clarify the taxonomic identity of *P. hopeiensis*. This study provides novel insight for the conservation management of *Pyrus* species and highlight the importance of plant taxonomy and phylogenomic analysis for biodiversity conservation.

## 2. Materials and Methods

### 2.1. Morphological Study

All specimens of *P. hopeiensis* (including type specimens) deposited in the herbarium of the Institute of Botany, Chinese Academy of Science (PE) and hundreds of specimens of

*P. ussuriensis* and *P. phaeocarpa* in PE were examined (Appendix A). Fourteen qualitative and quantitative characters were selected (Table 1), which included 1 binary and 13 continuous characters. Given differences in leaf morphology during the flowering and fruiting period in *Pyrus*, morphological analyses were performed for both periods. Aside from one specimen collected from an individual cultivated at PE (*Ren 2*, 21 August 1962), only specimens of *P. hopeiensis* identified by Yü and Ku were used to avoid misidentification. Two data matrices were made for 45 flower specimens and 61 fruit specimens, and each was treated as an operational taxonomic unit. Following Wang [32], the data matrix was standardized using a zero-mean normalization method. The formula $X^* = (X - m)/s$ was used, where "X" is the sample, "m" is the arithmetic mean, and "s" is the standard deviation. A principal coordinate analysis was performed based on the Gower general similarity coefficient analysis for mixed data sets using MVSP-Version 3.13b software.

**Table 1.** Morphological characteristics used in the principal coordinate analysis and their coding.

| | Specimens of the Flowering Period | | Specimens of the Fruiting Period |
|---|---|---|---|
| 1 | Leaf blade length [cm] | 1 | Leaf blade length [cm] |
| 2 | Leaf blade width [cm] | 2 | Leaf blade width [cm] |
| 3 | Ratio of leaf blade width to length | 3 | Ratio of leaf blade width to length |
| 4 | Leaf serration: (1) long; (0) short | 4 | Leaf serration: (1) long; (0) short |
| 5 | Length of flower stalk [cm] | 5 | Length of fruit [cm] |
| 6 | Indumentum density on stalk: (0) sparse; (1) dense | 6 | Persisting calyx: (0) no; (1) yes |
| | | 7 | Length of pedicel [cm] |
| | | 8 | Ratio of fruit to pedicel Length |

### 2.2. Phylogenomic Inference

In this study, whole-genome resequencing data of 27 samples of *Pyrus* representing 13 species from [33,34] were used for phylogenomic inference. Specifically, a total of 22 samples from Wu et al. [33] were employed, which included three individuals of *P. phaeocarpa*, five *P. ussuriensis*, and two *P. hopeiensis*. Five samples of *P. hopeiensis* (PWH8, PWH11, PWH18, PWH19, and PWH20) were used from Li et al. [34]. All samples were numbered following [33,34]. *Malus × domestica* L. was used as the outgroup, and its whole-genome sequence [35] was downloaded from NCBI.

Raw reads were filtered by removing several types of low-quality paired reads, including reads with adapters, paired reads with N content greater than 10%, and low-quality (Q < 10) paired reads contained in single-end sequencing reads that exceeded 50% of the length of the read. The accessions were then mapped to the apple genome using the MEM algorithm of Burrows–Wheeler Aligner in BWA software [36], and single nucleotide polymorphisms (SNPs) were called using the HaplotypeCaller module in GATK [37] and filtered using the following parameters: QD < 2.0‖MQ < 40.0‖FS > 60.0‖QUAL < −12.5‖ReadPosRankSum < −8.0—clusterSize 2—clusterWindowSize 5. The obtained SNPs were further filtered to construct a high-quality data matrix for phylogenetic inference with the minor allele frequency greater than 0.05, and less than 0.8 missing rates of the confirmed credible genotype from all accessions for each site and biallelic SNPs. Finally, a maximum likelihood (ML) phylogenetic tree was generated using IQTREE [38], and ultrafast bootstrap support values ($BS_{ML}$) were estimated with 1000 replicates [39].

A total of sixteen complete plastomes of *Pyrus* species were downloaded from NCBI, including one for each of the three species. *Malus kansuensis* (Batal.) Schneid. was used as the outgroup. Four samples of *P. hopeiensis* (PWH8, PWH11, PWH18, and PWH20) were selected from [34], one *P. ussuriensis* (1) was selected from [33]. The whole chloroplast genomes of these five samples were generated from the whole-genome resequencing data using the software GetOrganelle [40]. Plastomes of all samples were aligned by MAFFT [41]. After cutting both sides of the aligned sequences, a final data matrix with a length of 162,337 bp was obtained and used for phylogenetic analysis in IQTREE.

## 3. Results

### 3.1. Morphological Study

The results of principal coordinate analysis based on multiple characters showed that the four flower specimens of *P. hopeiensis* clustered with *P. ussuriensis* but were clearly separated from the *P. phaeocarpa* cluster (Figure 1A). Likewise, the six fruit specimens of *P. hopeiensis* were nested within the *P. ussuriensis* cluster, and this mixed group was distinct from that of *P. phaeocarpa* (Figure 1B).

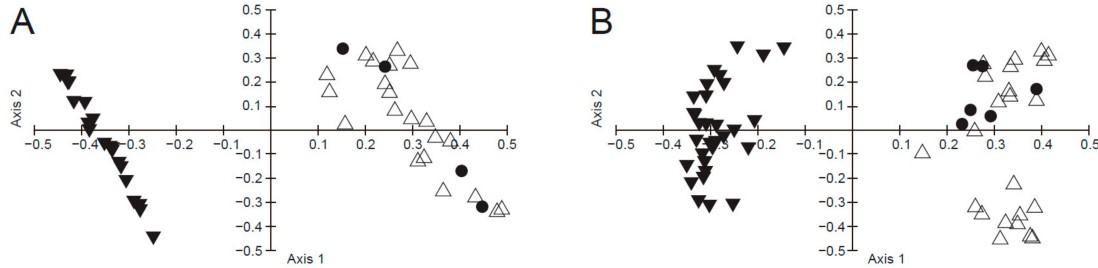

**Figure 1.** Principal coordinate analysis of *Pyrus hopeiensis*, *P. ussuriensis* and *P. phaeocarpa* based on morphological characters. (**A**) specimens of flowering period, (**B**) specimens of fruiting period. Black triangles: *P. phaeocarpa*, white triangles: *P. ussuriensis*, black circles: *P. hopeiensis*.

### 3.2. Phylogenomic Inference

After filtration, a total of 204.87 G clean data were obtained from whole-genome resequencing data, and a high-quality SNP data matrix with a length of 1,340,483 bp was generated. The ML tree was well resolved, and the seven samples of *P. hopeiensis* were divided into two groups. Two of them (PWH11 and PWH20) were grouped with five samples of *P. ussuriensis*, forming a monophyletic clade with full support (BS$_{ML}$ = 100, Figure 2A); this clade was sister to another fully supported monophyletic clade that comprised the rest of the *P. hopeiensis* samples (PWH8, PWH18, PWH19, 1 and 2). The three samples of *P. phaeocarpa* comprised a fully supported monophyletic clade (BS$_{ML}$ = 100, Figure 2A), and this clade was distinct from the *P. ussuriensis*–*P. hopeiensis* clade. Similar results were obtained in the plastome tree, with samples of *P. hopeiensis* and *P. ussuriensis* combined into a fully supported monophyletic clade (BS$_{ML}$ = 100, Figure 2B). A sample of *P. hopeiensis* from Shandong Province [28] was embedded within the subclade of *P. ussuriensis*. The clade of *P. phaeocarpa* was distinct from the *P. ussuriensis*–*P. hopeiensis* clade, and no phylogenetic discordance was detected among these three species.

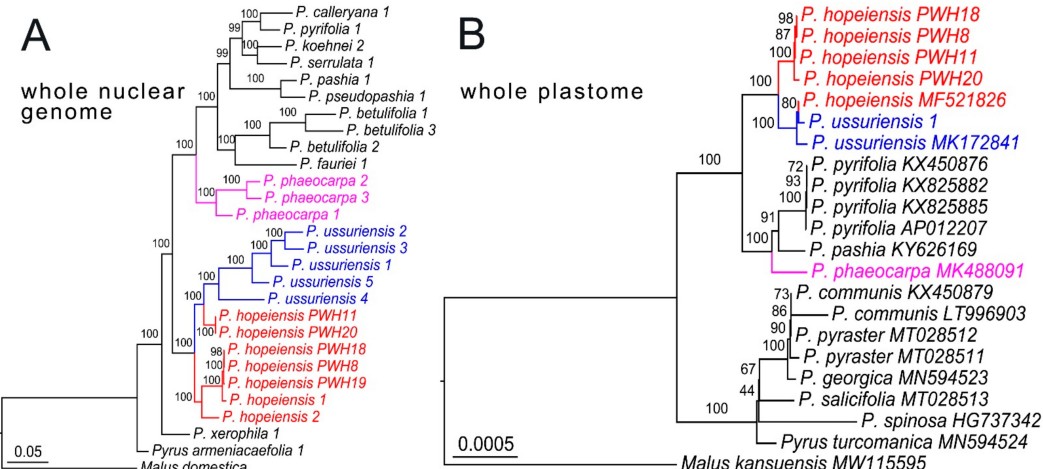

**Figure 2.** Maximum likelihood phylogenetic tree inferred from the SNP data matrix based on the whole-genome resequencing data (**A**) and whole plastid genome sequences (**B**). The numbers above

the nodes indicate bootstrap values generated from maximum likelihood analysis, scale bar indicates substitutions per site. Whole plastome sequences of species downloaded from NCBI are given their GenBank numbers following species' name in the plastome tree (**B**). Samples of *P. hopeiensis* are presented with red, *P. ussuriensis* with blue, and *P. phaeocarpa* with pink.

## 4. Discussion

### 4.1. Integrative Evidence for Reappraisal of the Identity of Pyrus hopeiensis

Morphological characters are important for species identification, and some morphological characteristics vary greatly among habitats and at different growth stages; this can result in inconsistencies in the designation of the taxonomic identity of species. The color of fruit and the spots on the surface of fruit are key morphological characteristics for discriminating between *P. hopeiensis* and *P. ussuriensis* [25]. The holotype (PE00020671, Figure 3A), isotype (PE00020670), and paratypes (*FS Chang 238, 269, 276*) of *P. hopeiensis* were collected in September 1953; this species was described 10 years later [25]. Although the fruit was described as brown in the original paper (Pomum . . . , fuscum, . . . ) [25] (p. 232), we found no fruit in these type specimens. However, the fruit is described as "mud-like yellow" (in Chinese) in the collection record of paratype *FS Chang 269* (Figure S1). Several brown or black fruits are present on two specimens collected in Shandong Province (*Zhou* et al. *1178, 1590*) on 6 and 7 June 1959 and identified as *P. hopeiensis* by Yü. Although "green fruit" is written clearly in Chinese on the record of *Zhou* et al. *1178*, these immature pomes turned black in herbaria (Figure S2). Hence, the fruit color of *P. hopeiensis* recorded in previous studies (brown) was not consistent with that of individuals observed in the field (yellow).

We conducted a long-term field investigation in North China since 2010. The fruit color of *P. ussuriensis* recorded in previous studies (yellow) is unstable. A dried fruit preserved in the lectotype of *P. ussuriensis* (Figure 3B) is pale brown. The color of mature fruit varies from greenish yellow to reddish yellow in the field (Figure 3C–E). One specimen collected by P. Zhang (*742*) on 13 September 1956 from Changli County, Hebei Province (Figure S3A) has a mature fruit that is reddish in color. Another specimen (*P. Zhang 766*, Figure S3B) presents three densely spotted fruits, and the fruit color (brown) is clearly indicated on the collection record (in Chinese). These two specimens were both identified by Yü as *P. hopeiensis* on 29 April 1960. The density of spots on the fruits both varies greatly within and among individuals; for example, there are several spots in Figure 3C (Dongling Mountain, in the north end of Taihang Mountain) and 3D (Yudu Mountain, in the west of Yan Mountain), but almost none in Figure 3E (Song Mountain, neighboring Yudu Mountain). Furthermore, persistent and caducous calyxes associated with mature fruits were observed in the same trees in the field (e.g., Figure 3B). Carpel number (4 vs. 5) and the length of the fruit pedicel (long vs. short) are sometimes used to differentiate *P. hopeiensis* from *P. ussuriensis*. However, our morphological analyses of specimens (Figure 1) and our field observations (Figure 3C–I) indicate that this trait has no taxonomic relevance.

We conducted two field investigations at the type locality—Jieshi Mountain, Xingshuyuan Village, Changli County, Hebei Province—at the end of August in 2015 and 2017. This is a small hill like the others located in the Yan Mountains: its altitude is 695 m, and it is 15 km from the Bohai Sea. Besides *P. hopeiensis* and *P. ussuriensis* documented by Li et al. [34], we found several individuals of *P. phaeocarpa* and *P. betulifolia*. The calyx was absent in mature fruit of these two species; fruits of the former are approximately 2 cm in diameter (vs. <1 cm in the latter), and the carpel number is 3–4 (vs. 2–3) (Figure 4). In 2015, we met an employee of the local forestry bureau who identified an individual as the endemic species, *P. hopeiensis* (Figure 4A). In Figure 4, fruits Aa, Bb, and C were collected from this individual. Key morphological characters (such as calyx on fruit, carpel number and leaf margin) indicate that this individual is instead *P. phaeocarpa*. Aside from a large number of wild *Pyrus* individuals, the Jieshi Mountain type locality includes a large orchard that contains *Pyrus* cultivars. One of the *P. betulifolia* trees growing here has fruits with an unusual green color (Figure 4B(c),D). This special germplasm resource has also been

discovered in Shanxi Province [42]. Unfortunately, this individual in the Jieshi Mountain died because of the growth of *Pueraria montana* (Loureiro) Merrill (Fabaceae) during our visit in 2017.

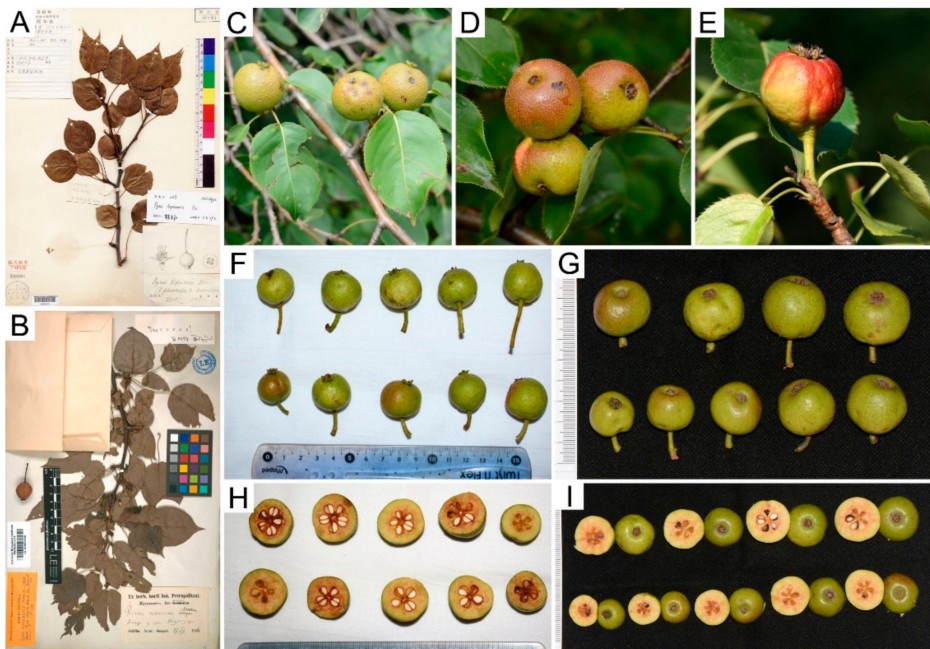

**Figure 3.** Type specimens of *Pyrus hopeiensis* and *P. ussuriensis* and fruit morphological characteristics of *P. ussuriensis* populations in North China. (**A**) Holotype of *P. hopeiensis*, (**B**) lectotype of *P. ussuriensis*, (**C–E**) fruit color, spots, and calyx, (**F,G**) fruit size, length of pedicel and calyx, (**H,I**) carpel number (4 vs. 5).

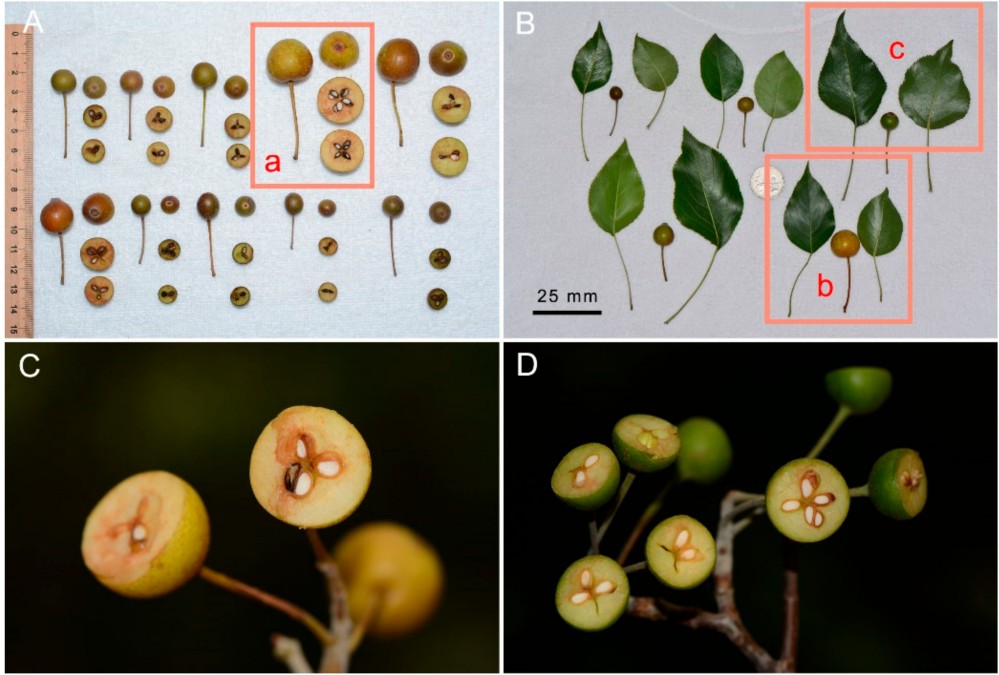

**Figure 4.** Fruits and leaves of *Pyrus phaeocarpa* and *P. betulifolia* from the type locality of *P. hopeiensis*, the Jieshi Mountain, Changli County, Hebei Province. (**A,B**) Fruits and leaves of *P. phaeocarpa* and *P. betulifolia*, (**C,D**) cross-cut of fruits of *P. phaeocarpa* and *P. betulifolia*, respectively, indicating 3–4 carpels per fruit. (**A(a),B(b),C**) were collected from the same tree of *P. phaeocarpa*, (**B(c),D**) were collected from the same tree of *P. betulifolia*.

Given the uncertain morphological differences between *P. hopeiensis* and *P. ussuriensis*, a phylogenetic study provides a powerful tool for clarifying the relationships between these taxa. However, the high level of genetic exchange and hybridization in Rosaceae at both the species and genus level [34,43–45] precludes the resolution of phylogenetic relationships among *Pyrus* species when traditional molecular markers are used, e.g., [12,19,46]. With whole-genome resequencing data, a robust molecular tree of *Pyrus* was constructed based on the SNP data matrix [33]. Two individuals of *P. hopeiensis* collected from the Zhengzhou Fruit Research Institute, Chinese Academy of Agricultural Sciences (CAAS), and Research Institute of Pomology, CAAS, were sampled. Both were embedded within a fully supported monophyletic clade of *P. ussuriensis*, and they were distinct from the *P. phaeocarpa* clade. We performed phylogenomic inference in this study based on the whole-genome resequencing data of Wu et al. [33] and Li et al. [34]. Our results demonstrate that all the *P. hopeiensis* samples were mixed with *P. ussuriensis*, and they together comprised one fully supported monophyletic clade (Figure 2). However, no cytonuclear discordance was detected for these three pear species between the nuclear and plastid genome data. Hence, a hybrid origin of *P. hopeiensis* from *P. ussuriensis* and *P. phaeocarpa* was not supported in this study. Both the co-existence of *P. ussuriensis* and *P. hopeiensis* at the same locations and their synchronous flowering and fruiting phenology suggest that there is no niche differentiation between them. Hence, evidence from morphological comparisons, phylogenomic inference, and our extensive field studies indicate that the locally endemic and protected *P. hopeiensis* is a synonym of the widely distributed *P. ussuriensis*. Consequently, 'the chosen one' of wild pear species (*P. hopeiensis*), which has received multiple conservation resources, should be excluded from the list of local key protected wild plants (Figure 5).

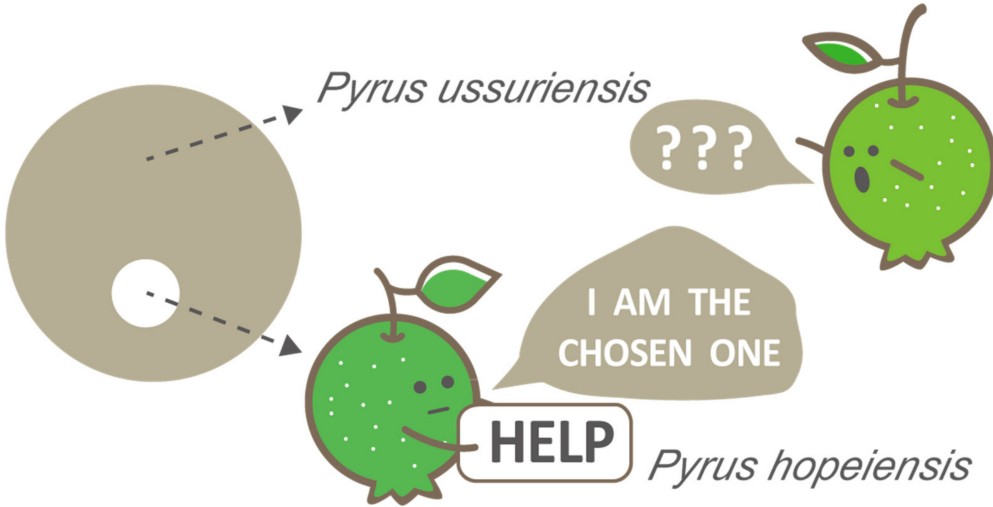

**Figure 5.** Integrative evidence indicates that the locally endemic *Pyrus hopeiensis* is a synonym of *P. ussuriensis* and should be excluded from the list of key protected wild plants.

*4.2. Integrative Evidence-Based Taxonomy for Biodiversity Conservation*

Climate change and biodiversity loss are two critical events of the Anthropocene, and the existence and distribution of biodiversity need to be better documented prior to proposing conservation strategies [47,48]. Although changes in the names of plants can be frustrating for conservationists, taxonomy is a science that involves classifying continuous variation in discrete categories and clarifying the phylogenetic relationships among taxa; taxonomists also have the responsibility of removing unnatural taxa that conflict with evolutionary and phylogenetic theory [49]. Species' names should change when a more rational and testable hypothesis of the species is produced based on more evidence [50]. For example, *Juglans hopeiensis* Hu (Juglandaceae), a species endemic to North China and listed as a key protected wild plant of Hebei Province, is thought to be a hybrid of *J. regia* L. and *J. mandshurica* Maxim. [51]. Whole plastome data suggested that it

was closely related to *J. mandshurica*, whereas the reduced-representation genomic data indicated a close relationship with *J. regia* [21]. Based on dense sampling whole-genome data, Zhang et al. [52] confirmed its hybrid identity and demonstrated that only the first generation of *J. hopeiensis* is viable, which precludes further speciation. On the one hand, the accurate identification of endangered species is a prerequisite before conservation action can be taken. On the other hand, species incorrectly listed as endangered species can result in a waste of valuable resources, and the prime of this are *P. hopeiensis* and *E. macrocarpa* [9]. Thus, integrative evidence is needed to assess the identity of controversial species, and these approaches should figure prominently in conservation management.

Because of limited resources, setting priorities for the conservation of threatened species is critically important. Species with unresolved taxonomies can pose problems for endangerment status assessments and impede conservation management; genetic data in the genomic era can provide information that can help clarify the taxonomic status of species [53]. In addition to living organisms, specimens preserved in museums and herbaria can provide valuable morphological and ecological data as well as genomic resources that can be used to resolve taxonomic uncertainties, reconstruct accurate phylogenies, and assess the mechanisms of ecological adaptation [54]. In some cases, type specimen may be the only credible remnant available for species identification. Hence, genetic resources from voucher specimens and even type specimens are vital for accurate identification, taxonomic designations, and conservation [55]. Although extracting DNA from museum specimens can be a major challenge and potentially be destructive [56–58], the growth of museomics, the study of DNA sequences obtained from museum specimens, combined with phenetic data collected from key specimens, will provide new taxonomic, evolutionary, and conservation insights.

*4.3. Taxonomic Treatment*

Based on the morphological comparison of specimens, phylogenomic inference, and our field studies, *P. hopeiensis* is a synonym of *P. ussuriensis*. The holotype of *P. ussuriensis* was not previously designated. Although several specimens are deposited in herbaria (e.g., GH, K and P), one of them preserved in LE that fit the original description of *P. ussuriensis* was designated as the lectotype.

*Pyrus ussuriensis* Maxim. ex. Rupr., Bull. Acad. Imp. Sci. Saint-Pétersbourg, sér. 2. 15: 132. 1857.—**Lectotype (designated here)**: Russia, *CJ Maximowicz s.n.* (LE, barcode LE01009639, [photo!]; isolectotype: LE barcode LE01009638, [photo!]). = *Pyrus hopeiensis* Yü, Acta Phytotax. Sin. 8: 232. 1963. *syn. nov.*—Type: China, Hebei Province, Changli County, Xingshuyuan village, 21 September 1953, *FS Chang 268* (PE barcode 00020671!; isolectotype: PE barcode 00020670!).

Other Specimens Examined

*Pyrus ussuriensis* Maxim. ex. Rupr.: **RUSSIA. Ussuri:** *CJ Maximowicz 96* (K barcode K000758073 [photo]), *CJ Maximowicz s.n.* (GH barcode GH00032508 [photo], LE barcode LE01009633 [photo], LE01009634 [photo]). **CHINA. Beijing:** Changping, 4 May 2001, *Changping Investigation Team 173* (PE barcode 01274862!), Haidian, June 1960, *Anonymous s.n.* (PE barcode 01274861!), 21 August 1962, *G Ren 2* (PE barcode 01608659!), Jiangou, 25 April 1959, *Hebei Investigation First Team 116* (PE barcode 00548756!), Mentougou, Xiaolongmen, July, 1978, *Beijing Normal University Team s.n.* (PE barcode 01274865!), Shangfangshan, 21 May 1959, *Hebei Investigation First Team 341* (PE barcode 00548750!); **Hebei Province**: Changli County, Xingshuyuan village, 1 Sept. 1953, *FS Chang 238* (PE barcode 00548753!), 21 Sept. 1953, *FS Chang 269* (PE barcode 00548751!), 24 Sept. 1953, *FS Chang 276* (PE barcode 00548755!), 13 September 1956, *P Zhang 742* (PE barcode 01448598!), Chicheng County, Zhenanpu town, 7 September 1959, *Anonymous 4100* (PE barcode 00548752!, 01274860!), Funing County, 24 April 1959, *Anonymous 58* (PE barcode 00548760!), Laiyuan County, 26 July 1959, *Anonymous 4-3743* (PE barcode 00548754!), Yi County, 24 July 1959, *Anonymous 3246* (PE barcode 01274859!), 25 July 1959, *Anonymous 3287* (PE barcode 01274858!),

Yu County, Xiaowutai Mountain, 27 August 1950, *YW Cui 2706* (PE barcode 01274864!), 28 August 1950, *YW Cui 2864* (PE barcode 01274863!); **Inner Mongolia Autonomous Region:** Zhaomengkaqi, Xiaoxigoumen, 4 June 1962, *Meng-Ning Comprehensive investigation Team 252* (PE barcode PE00548749!), 4 Aug. 1962, *Meng-Ning Comprehensive investigation Team 1292* (PE barcode PE00548748!); **Ningxia Autonomous Region:** Helan Mountain, 25 May 1959, *YQ He 02424* (PE barcode 00548759!); **Shandong Province:** Qingdao City, Laoshan, 7 June 1959, *TY Zhou* et al. *1590* (PE barcode 00548758!), 8 June 1959, *TY Zhou* et al. *1178* (PE barcode 00548757!); **Shanxi Province:** Xing County, *Yellow River Investigation Second Team 2695* (PE barcode 00548761!).

## 5. Conclusions

Plant names may change because of the development of taxonomic studies, which can aid biodiversity conservation. Because of frequent gene exchange and hybridization among species, the phylogenetic relationships among *Pyrus* species remain unclear, and the taxonomic identities of several species still require examination, such as *P. hopeiensis*, a regional key protected and potential hybrid species endemic to North China. Comprehensive morphological studies were performed based on flowering and fruiting specimens of *P. hopeiensis* and its potential parent, *P. ussuriensis*, and *P. phaeocarpa*. Extensive phylogenomic inferences were performed based on a high-quality SNP data matrix that was generated from whole-genome resequencing and the whole plastome data. A long-term field investigation of *Pyrus* species in North China was also conducted. Both morphological and phylogenomic studies indicate a close relationship between samples of *P. hopeiensis* and *P. ussuriensis*, and this mixed cluster consists of a fully supported monophyletic clade and is distinct from other species. Hence, *P. hopeiensis* is a synonym of *P. ussuriensis*. The lectotype of *P. ussuriensis* was designated here. Our work provides information for both the taxonomic study and conservation management of *Pyrus* and highlights the significance of integrative evidence-based taxonomy in biodiversity conservation.

**Supplementary Materials:** The following supporting information can be downloaded at: https://www.mdpi.com/article/10.3390/d14060417/s1, Figure S1. Paratype of *Pyrus hopeiensis*. *FS Chang 269*, PE barcode: 00548751. Figure S2. Specimen of *Pyrus hopeiensis*. *TY Zhou* et al., *1178*, PE barcode: 00548757. Figure S3. Specimens of *Pyrus hopeiensis*. A: *P Zhang 742* (PE barcode: 01448598), B: *P Zhang 766* (PE barcode: 01449638).

**Author Contributions:** X.-Y.M. conceived the work, performed field investigation, and prepared, wrote, and revised the manuscript; J.W. (Jiang Wu) carried out specimens' morphological statistics and the principal coordinate analysis; J.W. (Jun Wu) provided the original resequencing data of *Pyrus* species described in the manuscript. All authors have read and agreed to the published version of the manuscript.

**Funding:** This research was funded by the Natural Science Foundation of China (grant number 32070235), the Project of the Second Investigation of the National Key Protected Wild Plants of Beijing, and the Project of Agricultural Wild Plant Resources Investigation of Beijing (grant number 13200346).

**Institutional Review Board Statement:** Not applicable.

**Data Availability Statement:** All data are available in this article.

**Acknowledgments:** We thank Irina Illarionova (LE) for providing photos of the type specimen of *Pyrus ussuriensis*, and the curator and staff at PE for their help. We also thank Qin-Wen Lin, Yi-Xuan Zhu, Xue-Li Shen, Ling Tong, Wan-Jie Jiang, and De-Huai Zhang for their help during our field investigation.

**Conflicts of Interest:** The authors declare no conflict of interest.

## Appendix A

The information of specimens used for principal coordinate analysis given in the Appendix is taken from specimens deposited in PE.

*Pyrus hopeiensis* Yü: **CHINA. Beijing:** Changping District, 4 May 2001, *Changping Investigation Team 173* (PE barcode 01274862), Haidian, June 1960, *Anonymous s.n.* (PE barcode 01274861), 21 August 1962, *G Ren 2* (PE barcode 01608659), Jiangou, 25 April 1959, *Hebei Investigation First Team 116* (PE barcode 00548756), Mentougou, Xiaolongmen, July, 1978, *Beijing Normal University Team s.n.* (PE barcode 01274865); **Hebei Province:** Changli County, Xingshuyuan village, 13 September 1956, *P Zhang 742* (PE barcode 01448598), 19 September 1956, *P Zhang 766* (PE barcode 01449638), 22 September 1956, *P Zhang 782* (PE barcode 01449639, 01449640), Funing County, 24 April 1959, *Anonymous 58* (PE barcode 00548760).

*Pyrus phaeocarpa* Rehd.: **CHINA. Beijing:** Haidian District, Cheying village, 22 August 1972, *Anonymous Hai-676* (PE barcode 01204095), Mentougou District, Dongyangtuo village, 19 April 1954, *ZL Yan 167* (PE barcode 01448580), Mengwu village, 20 April 1954, *ZL Yan 171* (PE barcode 01448530); 20 August 1954, *ZL Yan 168* (PE barcode 01448527); **Hebei Province:** Changli County, Fenghuangshan, 21 April 1953, *TT Yü 35* (PE barcode 01448549, 01448550), near Railway Station Office, 27 April 1953, *TT Yü 092* (PE barcode 01448535), Xishan, *Anonymous s.n.* (PE barcode 01448554), Xingshuyuan, 17 April 1953, *TT Yü 13* (PE barcode 01448538), 20 April 1953, *TT Yü 30* (PE barcode 01448555), *TT Yü 227* (PE barcode 01448609, 01448611), West Zhanggezhuang, *Anonymous 14* (PE barcode 01448534), 16 September 1953, *Anonymous 6* (PE barcode 01448532), *Anonymous 9* (PE barcode 01448540), 24 April 1956, *P Zhang 647* (PE barcode 01448561, 01448562), 13 September 1956, *P Zhang 738* (PE barcode 01448567), *P Zhang 739* (PE barcode 01448560), *P Zhang 739* (PE barcode 01448569), *P Zhang 740* (PE barcode 01448569), *P Zhang 741* (PE barcode 01449634), Liugezhuang, 6 October 1953, *FS Chang 284* (PE barcode 00549049), Dongling, 22 April 1936, *DF Jin K-901* (PE barcode 00549045, 00549062), Funing County, Jiejia village, 22 September 1956, *FS Chang 776* (PE barcode 01448557, 01448558), *FS Chang 777* (PE barcode 01448559), *FS Chang 780* (PE barcode 01448575), Liugezhuang, 6 October 1953, *FS Chang 283* (PE barcode 00549046), 19 September 1956, *P Zhang 766* (PE barcode 01449638), Shibeigou, 24 April 1953, *TT Yü 58* (PE barcode 00549044), Yuanjiagou, 23 April 1953, *TT Yü 53* (PE barcode 01448543, 01448544), Qinghuangdao City, chitushan, 4 May 1953, *TT Yü 141* (PE barcode 01448545); **Henan Province:** Song County, 15 September 1956, *Bureau of Henan Forestry 986* (PE barcode 00549057), *Bureau of Henan Forestry 1259* (PE barcode 00549055, 00549058), 26 August 1959, *Anonymous 35149* (PE barcode 00549054), 1 October 1959, *Bureau of Henan Forestry 1259* (PE barcode 00549056); **Gansu Province:** Wushan County, 4 June 1956, *Yellow River Investigation Team 4480* (PE barcode 00549059); **Jiangxi Province:** Guangchang County, 7 October 1958, *QM Hu 5398* (PE barcode 00549061), Suichuan County, 22 September 1963, *JS Yue et al. 4168* (PE barcode 00549060); **Shaanxi Province:** Muhuguan, 22 June 1960, *Anonymous 0496* (PE barcode 01274917, 01274918), Shangzhou City, 4 July 1960, *Anonymous 1074* (PE barcode 01274921), Shanyang County, 12 May 1963, *ZY Zhang 15906* (PE barcode 01274919), 12 May 1964, *JX Yang 2656* (PE barcode 01274923), 22 May 1964, *JX Yang 2733* (PE barcode 01274920), 10 June 1964, *JX Yang 2929* (PE barcode 01274916), Banmiao, 6 May 1964, *JX Yang 2558* (PE barcode 01274922). **USA.** 10 May 1918, *HH Chung 4255* (PE barcode 01682167), 6 May 1930, *CEK and FPM 17501* (PE barcode 01682166).

*Pyrus ussuriensis* Maxim. ex Rupr.: **CHINA. Beijing:** Changping District, Nankou, 29 May 1956, *Herbarium Team 1198* (PE barcode 00549475), Fangshan District, 24 August 1956, *Herbarium Team 3529* (PE barcode 00549474), Haidian District, Xishan, Biyunsi, 22 April 1955, *Herbarium Team 561* (PE 00549442, 00549478), 10 July 1955, *Wofosi Investigation Team 153* (PE barcode 01448240), 1 August 1957, *YJ Zhang 327* (PE barcode 01448238), Western Park, 15 April 1951, *FZ Wang 11* (PE barcode 01449781), Mentougou District, Baihuashan, 22 July 1956, *CJ Liu and DY Xing 170* (PE barcode 00549483, 00549498), Dajuesi, April 1936, *DF Jin 12008* (PE barcode 01449776, 01449777), Zhoujiaxiang, April 1936, *Y Liu 12013* (PE barcode 01449775), 2 August 1936, *DF Jin 181* (PE barcode 01449774), *DF Jin 188* (PE 01449773), Miaofengshan, 23 May 1930, *HF Chow 40274* (PE barcode 00549481), 6 May 1953, *F Zhao 71* (PE barcode 00549506), Miyun County, Wulingshan, 4 May 1951, *Y Liu and J Zhang 15067* (PE barcode 00549448, 00549451, 00549487), Pinggu District, Beiyang Bridge,

23 May 1972, *Anonymous Ping-188* (PE barcode 01204098); **Hebei Province:** 1930, *HF Chow 40336* (PE barcode 00549496), 21 April 1936, *Anonymous 379* (PE barcode 00549493), Changli County, West Zhanggezhuang, 26 April 1956, *P Zhang 651* (PE barcode 01449433), Dingzhou City, Xinxingzhuang, 15 April 1956, *JY Zhong 529* (PE barcode 01449727), *JY Zhong 530* (PE barcode 01449735), Zhaozhuang, 22 May 1956, *JY Zhong 551* (PE barcode 01449737), Dongling, 23 April 1930, *HF Chow 40325* (PE barcode 00549479), 1 May 1932, *HF Chow 41931* (PE barcode 00549480), 22 April 1936, *DF Jin K-900* (PE barcode 00549445), 20 June 1956, *JX Duan 27* (PE barcode 00549441, 00549446), 25 June 1956, *JX Duan 180* (PE barcode 00549450), Funing County, 30 April 1956, *P Zhang 668* (PE barcode 01448400), Jiejiagou, 22 September 1956, *FS Chang 778* (PE barcode 01448651), Neiqiu County, Fuanmugou, *XY Liu and F Zhao 431* (PE barcode 01449778), Yu County, Xiaowutaishan, 13 September 1956, *Herbarium Team 2417* (PE barcode 00549476), Zhangjiakou City, 25 April 1956, *JX Duan 180* (PE barcode 00549447, 00549484, 00549485), Xiling, Huangtupo, 8 June 1953, *F Zhao 222* (PE barcode 00549488); **Tianjin:** Ji County, Panshan, 4 July 1956, *Herbarium Team 1941* (PE barcode 00549444); Precise locality unknown, *Anonymous 12049* (PE barcode 01449783), *Anonymous K-919* (PE barcode 01449782).

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
