# Peer review of "Taxonomic Uncertainty and Its Conservation Implications in Management, a Case from Pyrus hopeiensis (Rosaceae)"

_diversity, doi:10.3390/d14060417_

Round 1

Reviewer 1 Report

Since the sequences are taken from Ref. [27], lines 105-115 may be deleted. Instead of this indicate how many SNPs are located in coding and non-coding regions and % of synonymous non-synonymous substitutions

The word ”Nested” means placed one inside the other.Change “nested” to “grouped” or “combined”:

Lines 143-144 Two of them (PWH11 an PWH20) were nested with five samples of P. ussuriensis.

Line 149 “samples of P. hopeiensis and P. ussuriensis nested together”

Ref [34] does not concern data on “complete sequence of chloroplast genome, as well as [35] is not about hybridization.

Reviewer 2 Report

The study provides phylogenomic evidence that P. hopeiensis, thought to be a hybrid of P. ussuriensis and P. phaeocarpa and listed as an endangered species, is a synonym of P. ussuriensis. The information is potentially useful for defining conservation management policies and priorities.  I do not see problems with the experimental approach. The manuscript would benefit from formal refinement and streamlining. I added suggestions in the text (see pdf attached), but there are some points that need being revised by the authors.

Lines 82-83: All specimens of P. hopeiensis (including types and normal specimens deposited in

  1. PE) and several specimens of P. ussuriensis and P. phaeocarpa were examined. What do the authors mean by “all”? What is “PE”?

Line 87: what do the authors mean by “sheet”

Line 153: no phylogenetic discordance was detected among these three species. Explain

Lines 217-218: this individual died because of the growth of Pueraria montana (Loureiro) Merrill

(Fabaceae) during our visit in 2017. Which individual, the one in the Hebei Province or that in the Shanxi Province?

Lines 235-236: Recently, Li et al. [28] conducted a field study in Jieshi Mountain  and re-sequenced several Pyrus individuals, which included five samples of P. hopeiensis. This sentence appears to be out of place here. Modify it in order to justify its presence; alternatively, move it to a better place or delete.

Lines 240-241: A hybrid species usually nest with one of its parents on trees of the plastome and nuclear genome, respectively. The sentence is rather obscure. Explain.

Lines 243-244: insert reference

Lines 252-253: and we need to better document the existence and distribution of biodiversity when direct conservation strategies. The sentence is obscure. Reword it appropriately.

Lines 338-339: We also suggest museomics for taxonomic ambiguities which may be scheduled for conservation management. The sentence is almost senseless. Reword or delete.

Reviewer 3 Report

The Rosaceae family includes many genera, the taxonomy of which is controversial due to the large and poorly recognized range of species' phenotypic plasticity. Wild species of Pyrus are widespread in Europe, North Africa, west, central and east Asia and Japan. These plants hybridize easily, both with cultivated and wild species. Consequently, a considerable number of intermediate forms and more or less stable segregants has arisen causing obvious taxonomic difficulties. The aim of the present work was to carry out an integrative taxonomic study, based on the results of both morphological and phylogenomic studies, in order to establish the position and status of the enigmatic species Pyrus hopeiensis. In this work, the authors conducted a comprehensive morphological study of three species, i.e. P. hopeiensis, P. phaeocarpa and P. ussuriensis, and performed a phylogenomic analysis based on whole-genome resequencing data and complete chloroplast genomes. This manuscript is in general well written, logically structured, well-illustrated and easy to understand. It also addresses a subject that is of great interest in the scientific community. The title clearly describes the contents of the paper. The abstract is well written. The introduction is well written as it gives a good background of the research in question. Also, the aim of the study is evident in the beginning and concluding parts. I believe that the Materials and Methods section is well structured and scientifically sound. The results are well presented, figures and tables are correct. Literature reviews in the discussion section of the manuscript are very professional. My comments mostly relate to relatively minor issues of interpretation and writing. These comments do not influence a positive impression of the article.

Suggestions:

Line 47: More than 80 species are recognized within the genus Pyrus (Browicz, 1993) and 73 of them are abundant in Eurasia (Robertson et al., 1991).

Reference: Browicz K. 1993. Conspect and chorology of the genus Pyrus L. Arboretum Kórnickie 38: 17-33. [in Polish].

Robertson K.R., Rohrer J.R., Rohrer J.B., Smith P.G. 1991. A synopsis of genera in subfamily Maloideae (Rosaceae). Syst Bot, 16: 376–394.

Line 82: specify what "normal specimens" means, do you mean representative?

Line 83: several - how many specimens? please specify

Table 1: Specimens of the flowering period – do you mean at the flower blooming stage?

Reviewer 4 Report

The authors tried to discuss the conservation managements from the taxonomic uncertainty of Pyrus. Of course, it’s a novel idea and suitable to be published as a possible publication after revise. There is also a paper from taxonomy to conservation, Lian et al., 2021, Taxon, 70, 931-945. Pls check this and some viewpoints can be discussed in your manuscript. Also, many improvements are necessary in the revised manuscript.

The specific issues I concern are:

Title is too long, and I suggested “Taxonomic uncertainty and its conservation implications in management, a case from Pyrus hopeiensis (Rosaceae)”;

Line 17, an enigmatic species endemic to North China, need to be rephrased;

Line 25-27, need to be rephrased;

Line 42, the logic relationship from here to the next section, should add sth. to link the pears, such as the mistakes/errors in identification, then influence the conservation unity etc., Goodwin et al., 2015, Current Biology, 25, R1077-1067.

Line 44, pls say more about the taxonomy of Pyrus and the status, and then infer the taxonomic uncertainty.

Line 55 “an enigmatic species endemic to North China”, rephrase this sentence;

Line 58-72, rephrase this section, should be more coherent, and clear logic;

Line 74-79, pls propose your scientific issues with power, not so weak and feeble in current version;

Line 83 “several specimens of P. ussuriensis and P. phaeocarpa were examined.” Where are these? Also PE or another locality?

Line 88 “identified by Yü and Ku were”, if you consider the misidentification, pls give the right identification here, such as, should be sth.

Line 99 “whole-genome resequencing data of 27 samples of”, should be pointed the sources of WGS of the 27 sample here.

Figure 1 and the other figure 2, pls provide high resolution pictures.

Line 98-99, move this sentence to Introduction section;

Line 158, you mentioned the GenBank accession nos. are presented following the name, but A and few in B without the accession nos., pls provide them.

Line 169, “(Pomum …, fuscum, …)”, what’s mean of this?

Line 171, 175, the mentioned Figure S1 and S2, where, pls provide these in appendix, and the Appendix should be in another part and in the web version when it is published;

Lin 251-253, rephrase this sentence;

Line 257-268, so many references here, but just one sentence in 269-270 you want to tell, it is a pity if just so simple evidence you want to explore here. So this section should be improved;

Line 271-272, is it true for the limited financial support and resources???

Line 295 “[photo!];”, where, or what the mean of ! here?

Line 298, “!”? and the others in the following section,pls check these;

Line 324, in the Conclusion section, I think I have gotten what you want to explain, but I suggest you can give more information, including the viewpoints or some perspective of this topic; thus, I suggest you rephrase this section, and make your viewpoints clearer.

Round 2

Reviewer 4 Report

I appreciate all improvements from the authors, and the current form can be considered as publication in Diversity.

But, the fugures 1, and 2 are still not clear, I suggest the authors provide high-resolution for final publication.